# Optimal Woven EndoBridge (WEB) Device Size Selection Using Automated Volumetric Software

**DOI:** 10.3390/brainsci11070901

**Published:** 2021-07-08

**Authors:** Sameer Ansari, Cynthia B. Zevallos, Mudassir Farooqui, Andres Dajles, Sebastian Schafer, Darko Quispe-Orozco, Alan Mendez-Ruiz, Samir Abdelkarim, Sudeepta Dandapat, Santiago Ortega-Gutierrez

**Affiliations:** 1Department of Neurology, University of Iowa Hospitals and Clinics, Iowa City, IA 52242, USA; sameer-ansari@uiowa.edu (S.A.); cynthia-zevallos@uiowa.edu (C.B.Z.); mudassir-farooqui@uiowa.edu (M.F.); andres-dajles@uiowa.edu (A.D.); darko-quispeorozco@uiowa.edu (D.Q.-O.); alan-mendezruiz@uiowa.edu (A.M.-R.); samir-abdelkarim@uiowa.edu (S.A.); sudeepta.dandapat@aah.org (S.D.); 2Siemens Healthcare, Imaging and Therapy Systems, 91301 Forchheim, Germany; schafer.sebastian@siemens-healthineers.com; 3Department of Radiology, University of Iowa Hospitals and Clinics, Iowa City, IA 52242, USA; 4Department of Neurosurgery, University of Iowa Hospitals and Clinics, Iowa City, IA 52242, USA

**Keywords:** Woven EndoBridge, WEB device, aneurysm, embolization, endovascular

## Abstract

Introduction: Selecting the appropriate Woven EndoBridge (WEB) device sizing for the treatment of wide-neck bifurcation aneurysms (WNBAs) remains challenging. The aim of this study was to evaluate different volumetric-based imaging methodologies to predict an accurate WEB device size selection to result in a successful implantation. Methods: All consecutive patients treated with WEB devices for intracranial aneurysms from January 2019 to June 2020 were included. Aneurysm dimensions to calculate aneurysm volumes were measured using three different modalities: automated three-dimensional (3D) digital subtraction angiography (DSA), manual 3D DSA, and two-dimensional (2D) DSA. The device–aneurysm volume (DAV) ratio was defined as device volume divided by the aneurysm volume. WEB volumes and the DAV ratios were used for assessing the device implantation success and follow-up angiographic outcomes at six months. Pearson correlation, Wilcoxon Rank Sum test, and density approximations were used for estimating the WEB volumes and the imaging modality volumes for successful implantation. Results: A total of 41 patients with 43 aneurysms were included in the study. WEB device and aneurysm volume correlation coefficient was highest for 3D automatic (r = 0.943), followed by 3D manual (r = 0.919), and 2D DSA (r = 0.882) measurements. Measured median volumes were significantly different for 3D automatic and 2D DSA (*p* = 0.017). The highest rate of successful implantation (87.5%) was between 0.6 and 0.8 DAV ratio. Conclusion: Pre-procedural assessment of DAV ratios may increase WEB device implantation success. Our results suggest that volumetric measurements, especially using automated 3D volumes of the aneurysms, can assist in accurate WEB device size selection.

## 1. Introduction

Approximately 26–36% of intracranial aneurysms (IAs) are wide-neck bifurcation aneurysms (WNBAs), defined as aneurysms with neck diameters ≥4 mm and a dome-to-neck ratio of ≤2 mm [1,2]. Endovascular treatment of WNBAs is challenging because it requires embolization of the aneurysm while preserving the bifurcation vessels to achieve complete occlusion [1]. With the recent advancement in minimally invasive endovascular treatments, several devices and techniques have been used, including balloon-assisted coiling, stent-assisted coiling (SAC), flow diverters, and flow disruptors. The latter have recently become popular due to their safety and efficacy [3]. The Woven EndoBridge (WEB, MicroVention, Inc., Aliso Viejo, CA, USA) is the first intrasaccular flow disruption device approved by the United States (US) Food and Drug Administration (FDA) specifically for the treatment of saccular WNBAs [3,4].

The WEB is a self-expanding microbraid and nitinol wiring device, with the highest density of wires located at the base [1]. Once the device is placed into the aneurysm, it expands into a sphere (single layer sphere, SLS) or a cylinder (single layer, SL), obliterating the flow within the aneurysm causing thrombosis [1]. Unlike the SAC technique, which is most commonly used for these aneurysms, the WEB device does not require dual anti-platelet therapy, which decreases the risk of hemorrhagic complications associated with ruptured aneurysms [3]. Moreover, it has shorter anesthesia and procedural time metrics, and may be a more cost-effective treatment procedure [5]. Despite multiple advantages, successful implantation is sometimes challenging, resulting in either replacement of the device and/or an addition of stents/coils during the procedure. Moreover, these WEB devices can sometimes compress over time, resulting in aneurysm recurrences that require retreatment [6].

The selection of optimal WEB device size can be a major factor determining the success of implantation. Oversized or undersized devices may result in thrombosis of the parent vessels or inadequate aneurysm obliteration, respectively [1]. Imaging modalities for measuring aneurysm dimensions (diameter, height, and neck width) include computed tomography (CT) and magnetic resonance (MR) angiography (usually used as first stage, diagnostic tools) as well as two-dimensional (2D) and three-dimensional (3D) digital subtraction angiography (DSA) [1]. In current practice, a 3D DSA reconstruction and/or 2D DSA is used to determine the ideal WEB device size [1]. It is recommended to oversize the device (in comparison with the diameter of the aneurysm) to achieve an adequate radial force across the surface and neck coverage. The size is selected by adding 1–2 mm (1 mm for smaller and 2 mm for larger aneurysms) to the average width and subtracting the similar amount (millimeters) from the height of the aneurysm [1]. However, even with these guidelines, successful implantation is not guaranteed.

Studies have previously described volume embolic ratio (VER), which measures the volume of coils inside the aneurysm, to accurately estimate the size of the device needed and to predict aneurysm recanalization [7,8,9]. In the interest of enhancing WEB device implantation, we evaluated DSA imaging methodologies that allow accurate WEB device size selection using volumetric measurements. Specifically, we aimed to better characterize the optimal device–aneurysm volume (DAV) ratio to predict successful WEB device implantation.

## 2. Methods

We conducted a retrospective observational cohort study of all consecutive patients with intracranial aneurysms who underwent treatment with a WEB (SLS or SL) device at our comprehensive stroke center from January 2019 to June 2020. The aneurysms were selected following the WEB Intrasaccular Therapy (WEB-IT) Study characteristic’s requirements [3]. Patient demographics, clinical and procedural data, along with device and aneurysm measurements at the initial treatment and at follow-up, were collected.

Clinical variables included medical history, symptoms at presentation, aneurysm location, rupture status, and use of any antiplatelet medications. Procedural variables included number of devices, implantation attempts, any periprocedural stroke and/or hemorrhage, use of adjunct coil or stent, and any additional procedure. Aneurysm measurements included height, width, depth, neck, and volume. All neuroimaging data from DSA sources was obtained from two independent investigators not involved in the treatments (CBZ, DQO). When disagreement occurred, cases were resolved by a third senior neurointerventionalist, when needed (SOG). This study was approved under the waiver of informed consent by the local institutional review board, and it is reported in accordance with the Strengthening the Reporting of Observational Studies in Epidemiology (STROBE) guidelines [10].

### 2.1. Aneurysm Measurement

Aneurysm dimensions were measured using three different modalities including automated 3D DSA, manual 3D DSA, and 2D DSA (Figure 1). Automated pretreatment 3D DSA volumes (method 1) were measured using proprietary aneurysm analysis software (Aneurysm Analysis, Siemens Healthineers, Forchheim, Germany). Reconstructions performed used a vessel enhancing kernel and a smooth image impression focused to minimize image noise, a potential detriment to the automatic analysis. Dimensions (height, width, neck, and volume) were automatically calculated after placing seed pixels, one point in the center of the aneurysm sac and two points in the parent vessel (proximal and distal). The tool applies an automatic thresholding to separate the vessel from background information (Figure 1A–C). This software was used due to its reported accuracy for aneurysm measurements as compared to other automatic modalities [11,12].

On the other hand, manual calculations were done in both pretreatment 3D DSA reconstruction (method 2), and 2D DSA (method 3) with anterior-posterior and lateral planes. Automatic 2D image calibration was used to determine 2D DSA measurements, and the reported tool error margin was 1.5% (Syngo X-Workplace, Siemens Healthineers, Forchheim, Germany). The aneurysm height was measured perpendicular to the neck, and the width was measured parallel to the neck at the widest point of the aneurysm. The depth was measured perpendicular to the height in the lateral plane. Manual aneurysm measurements were used to approximate aneurysm volume by using the equation π/6 × width × depth × height, under the assumption that the aneurysms had an ellipsoidal body [13,14] (Figure 1D–F).

### 2.2. Device–Aneurysm Ratio

The DAV ratio was calculated to evaluate the association with the immediate successful implantation and aneurysm obliteration at follow-up. DAV ratio is defined as the volume of the WEB device divided by the volume of the aneurysm. WEB device volume was calculated for the spherical (SLS) and cylindrical (SL) devices using height and width measurements provided by the manufacturer.

### 2.3. Radiological Outcomes

Primary and secondary outcomes were to evaluate the DAV association with the immediate successful implantation and aneurysm obliteration at follow-up, respectively. Immediate successful implantation of the device was considered if the device was completely introduced into the aneurysm without protrusion into the parent vessel. Unsuccessful implantation was considered if the device protruded, if coils or stents were used in conjunction with the WEB device, or if the operator changed the device based on inaccurate WEB device size. Each attempt was considered separately for the analysis.

Aneurysm obliteration at 6 month follow up was evaluated using the WEB-IT and Raymond Roy (RR) classifications [3,15]. WEB-IT is classified as the following: Grade A, no contrast in neck or sac of aneurysm; Grade B, no contrast in aneurysm sac but some contrast in neck of aneurysm; Grade C, contrast in aneurysm neck and sac. RR classifications are the following: RR 1, complete obliteration; RR 2, residual aneurysm; RR 3a, residual aneurysm with contrast inside of the embolic device; RR 3b, residual aneurysm with contrast around the aneurysm sac [3,15]. The radiographic follow-up outcomes were dichotomized as successful and unsuccessful based on RR 1, 2 versus 3a, 3b and WEB-IT Grades A, B versus C, respectively.

### 2.4. Statistical Analysis

Categorical variables are reported as integers and percentages. A correlation between all the WEB device volumes and the three imaging modality volumes (3D automatic, 3D manual, 2D DSA) was performed to evaluate the accuracy of the 3 imaging modalities using a Pearson correlation coefficient. Because of the uncertainty in the estimation of the correlation coefficient that comes from the small sample size, a 1000 bootstrap sample was used to construct a distribution of the correlation coefficients for each imaging modality. The bootstrap distribution provides an estimate of the distribution of coefficient values in the general population of interest. In addition, a Fisher r-to-z transformation was used for the correlation coefficients to access any statistically significant differences.

WEB volumes and the imaging modality volumes were used to calculate the DAV ratios. To analyze the ratios among different aneurysm sizes, we conducted a two-way ANOVA between aneurysms of diameter less than 5 mm (n = 7), ≥5–7 mm (n = 13), and >7 mm (n = 9).

Using a density approximation, these ratios were then plotted for the successful and unsuccessful implantations. Finally, box plots of DAV ratios versus the obliteration of both RR and WEB-IT classifications at follow-up were plotted. The Wilcoxon rank sum exact test was used for calculating any statistical significance between the two groups. A *p*-value of ≤0.05 was considered statistically significant. All the analyses were performed using R software for Windows, version 3.5.2 (R Foundation for Statistical Computing, Vienna, Austria).

## 3. Results

A total of 41 patients with 43 aneurysms were included in this study. Clinical and procedural characteristics are summarized in Table 1. Most of the aneurysms had an ellipsoid/spherical shape, and 8 were considered having irregular shape with 2 or more daughter sacs/lobes. Of these 43 aneurysms, 29 underwent successful implantation while 14 aneurysms had unsuccessful implantation on their first attempt. Of the unsuccessful implantation group, 7/14 had an additional stent and/or coil placed, whereas the other half had the first WEB device removed and replaced by a newly sized WEB device. Six out of 7 of these second attempts resulted in success, while 1/7 resulted in failure and had to be stented (Appendix A). Of these 43 first WEB attempts, and 7 second WEB attempts, 19 successful implantations and 10 unsuccessful implantations had 3D reconstruction that was used for analysis.

The correlation between WEB device volume and the aneurysm volume calculations (Figure 2) showed that all three modalities were highly correlated. The 3D automatic modality (r = 0.943) was the most accurate, followed by 3D manual (r = 0.919), and 2D DSA (r = 0.882). The 95% confidence intervals for each bootstrap distribution were (0.87, 0.98), (0.82, 0.97), and (0.89, 0.99) for 3D automatic, 3D manual, and 2D DSA, respectively. Moreover, our results indicate that the volume measured using 3D automatic was similar to 3D manual (*p* = 0.26), whereas it was significantly different when compared with the 2D DSA (*p* = 0.017).

The density approximation graph shows the conditional ratio probabilities of success (Figure 3). Concretely, the 3D automatic data of the DAV ratios that resulted in successful implantation have a distinct range of ratio values that were associated with higher success with an upward bound of 1. The highest probability of successful implantation occurs between 0.6 and 0.8 DAV ratio with a probability of success of 87.5%. Ratios smaller or equal to 0.6 have a 66.7% probability of success, whereas with ratios greater than 0.8 the probability of success is 44.4% (Figure 3A).

On the other hand, many of the DAV ratios that resulted in success from the 3D manual and 2D DSA are above 1, and some DAV ratios for 2D DSA are above 2 (Figure 3B,C). This suggested that the WEB device is much larger than the aneurysm dimensions, thereby physically protruding into the parent vessel. Moreover, the observations from the density approximations in our study were not able to identify a distinctive differentiation of DAV ratios among the successful and unsuccessful WEB device implantations (Figure 3B,C). There were no significant differences in the mean ratios among the different size groups (*p* = 0.164), or with regards to their successful or failed implantation status (*p* = 0.105).

Finally, we evaluated the relationship between DAV ratios measured by 3D automatic calculation and radiographic outcomes in 17 patients who had follow-up imaging. The box and whisker plots display that there were no statistical differences between successful and unsuccessful obliteration groups regarding the DAV ratios; RR 1, 2 ratios (median 0.693, interquartile range [IQR] 0.572–0.812) versus RR 3a, 3b ratios (median 0.696, IQR 0.647–0.711) (*p* = 0.59) (Appendix A). Results were mirrored when using WEB-IT classification.

## 4. Discussion

In this study, the automatic aneurysm volume calculations using a proprietary aneurysm analysis software (Aneurysm Analysis, Siemens Healthineers, Forchheim, Germany) demonstrated the highest correlation with the WEB device volumes, especially when compared to manually calculated volumes using 3D and 2D imaging. When using this software, we were able to identify an optimal DAV ratio for successful implantation between 0.6 and 0.8.

Appropriate WEB device size selection is of utmost importance for successful treatment. It is recommended that the width of the WEB device should be marginally oversized relative to the diameter of the aneurysm, so that the radial force exerted by the WEB can help brace itself against the aneurysm walls and bridge the neck completely [1]. In the current practice, proceduralists choose the WEB devices based on the manufacturer’s recommended sizing chart, using 2D measurements of aneurysm height and width [1]. However, even with these guidelines, appropriate device sizing is challenging, requiring the opening of multiple devices, and technical success is not guaranteed. In our study, we observed that the automatic 3D measurement acquisition might outperform 3D manual and 2D DSA volumetric measurements (Figure 3). Although both 3D automatic (r = 0.943) and 2D DSA (r = 0.882) highly correlate with WEB device volume measurements, the two modalities were significantly different (*p* = 0.0167), suggesting that the 3D measurements are more accurate than the 2D DSA. The 3D image analysis allows measurements from multiple planes, providing more information than 2D images. 3D image acquisition also provides spatial orientation of aneurysm and surrounding vasculature, which may improve measurement accuracy. Furthermore, intracranial aneurysms are inherently 3D structures, thus measuring by only using height and width dimensions may not provide an accurate size of the aneurysm. Our results indicate that 3D volume measurements, providing more accurate impression of the aneurysm size, could be used for appropriate WEB device selection.

With current advancement in the technology, studies have shown that automated software analysis can help interventionists in better planning and decision-making strategies [16,17]. Recently, Cagnazzo et al. described their experience of utilizing automated software for predicting the optimum WEB device selection by reconstructing an aneurysm from 3D rotational angiography [18]. They observed that the use of automated software reduces the time of procedure, radiation dose, incorrect device selection, and multiple interventions. Previously, similar results were observed with pipeline devices for aneurysm embolization [17]. Nevertheless, with the increasing utilization of automated WEB device optimization protocols, it is imperative to prospectively evaluate technical feasibility and efficacy for the guidance of optimal treatment strategies and management.

We also observed that the highest rate of successful WEB device implantation was between 0.6 and 0.8 DAV ratio. Previous studies evaluating aneurysm coils have described volume embolization ratio (VER), or coil volume divided by aneurysm volume, as a predictor of aneurysm recanalization. A study by Neki et al. reported a VER value of 33.0% to avoid recanalization of the aneurysm, while other authors have reported a similar VER range of 18–31%, whereas low VER and high VER are risk factors for recanalization after coil embolization and coil migration, respectively [7,9,19,20]. Our results demonstrate the ideal volumetric range for the WEB device inside the aneurysm for acquiring optimal technical success, suggesting that both undersized and oversized WEB devices may result in technical inaccuracies. Smaller devices insufficiently seal the neck of the aneurysm, allowing blood to flow around the device and into the aneurysm. Similarly, oversized devices can protrude into the parent vessel and result in occlusion and/or thrombosis. This technical failure results in either retracting and replacing the device or using an additional embolic device such as stent(s) or coil(s). Moreover, use of these adjunct devices may increase the risk of perforation and peri-procedural and/or long-term complications. Thus, inadequate measurements and WEB device sizing may lead to procedural complications, technical WEB device implantation failure, incomplete occlusion rates, and ineffective treatment. When we analyzed the DAV ratios among the different aneurysm sizes, we found no significant differences regarding their implantation status.

Our analysis showed a DAV ratio of 0.6–0.8 to have highest rates of technical success. However, this was not reflected in the radiographic follow-up outcome. Instead, there was no statistical significance for the DAV ratios and aneurysm obliteration. This may be because the DAV ratio can change after implantation due to change in the shape of the WEB device (referred to as compression or compaction) [1], (Appendix A). However, the loss of follow-up decreased our sample size substantially (N = 17), rendering the study to be underpowered for conclusive inferences in these groups [1,6]. Change in WEB device height alters the volume and hence the DAV ratio. While some studies observed similar results reporting a non-significant relationship between the WEB device size selection and compression at follow-up [21,22], others reported specific factors such as aneurysm morphology, size and/or increased arterial blood flow to be associated with WEB device shape modification [21,22,23]. In our study, only eight aneurysms were found to have irregular margins with multiple daughter sacs/lobes. These aneurysms were small and were considered during the initial measurements for manual volume calculations. Furthermore, the automatic software accounted for the volume calculation, thereby providing accurate measurements for these aneurysms.

This study has some limitations. It is a single center-study with retrospective analysis of the data and a small sample size. We performed bootstrapping to account for the limited number of volumes, and the resulting narrow intervals indicated that the correlations between imaging modalities and device volumes were high despite the randomness in the sample used. We could not find an association between DAV ratios and aneurysm obliteration at follow up, but this was not the main focus of our study, as our aim was to propose a novel sizing modality for WEB devices. Larger sized, prospective studies may be needed in this regard.

## 5. Conclusions

WEB devices have been demonstrated as a safe and effective treatment modality for intracranial aneurysms [24]. DAV ratio using an automatic 3D software is an important factor that might help enhance technical success of WEB endovascular procedures by improving the selection and accuracy of device sizing. Prospective studies to evaluate the effectiveness and feasibility of this modality are needed before considering its implementation.

## Figures and Tables

**Figure 1 brainsci-11-00901-f001:**
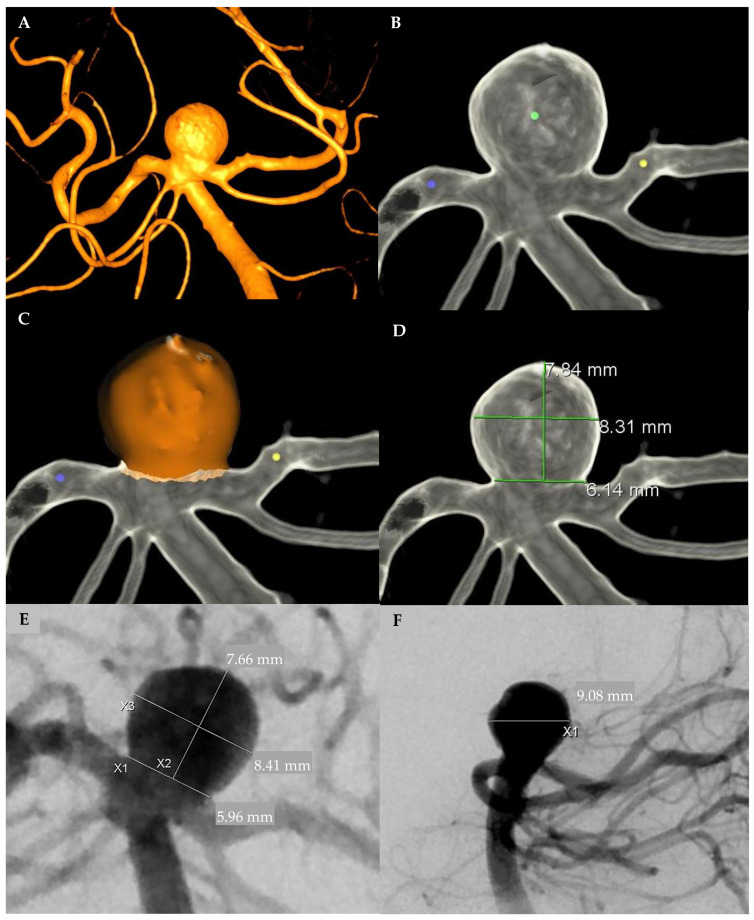
Angiographic measurement modalities. (**A**) Three-dimensional (3D) angiographic reconstruction of the aneurysm. (**B**) Automated 3D: Seed pixels within the aneurysm and the proximal and distal parent vessels. (**C**) Automated 3D: Automatic segmentation analysis and calculation of the volume of the aneurysm (400 mm^3^). (**D**) Manual 3D: manual measure of height, width, and neck of aneurysm. (**E**) Manual 2D—anteroposterior view: manual measure of height, width, and neck of aneurysm. (**F**) Manual 2D—lateral view: manual measure of depth.

**Figure 2 brainsci-11-00901-f002:**
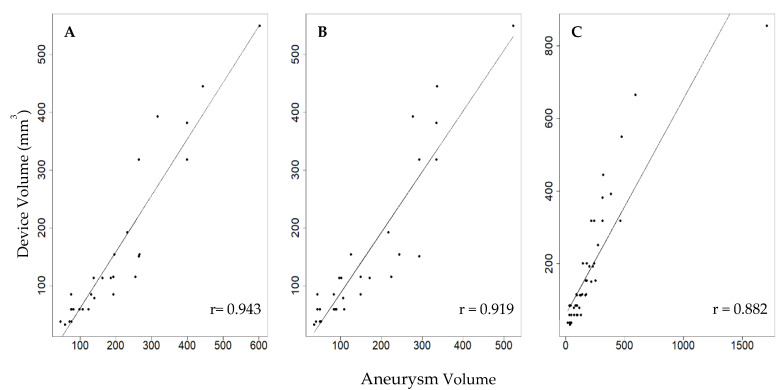
Correlation between WEB device volume and aneurysm volume measurements using (**A**) 3D Automatic (n = 29), (**B**) 3D Manual (n = 29), and (**C**) 2D Manual DSA (n = 48).

**Figure 3 brainsci-11-00901-f003:**
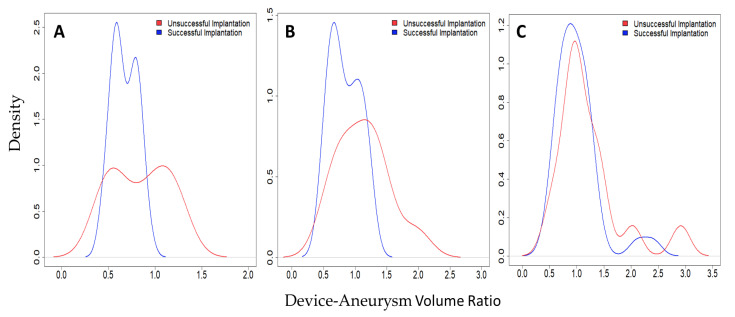
Density approximations of DAV ratios measured by using (**A**) 3D automatic, (**B**) 3D manual, and (**C**) 2D manual DSA. Successful implantation ratios appear in blue; Unsuccessful implantation ratios appear in red.

**Table 1 brainsci-11-00901-t001:** Demographics and characteristics of patients with WEB device treatment.

Variable	Number
No. of Patients (aneurysms)	41 (43)
Proportion of Women	33/43 (77%)
Median Age	66 years
Past Medical History	
Hypertension	24/43 (56%)
Diabetes	3/43 (7%)
Atrial Fibrillation	3/43 (7%)
Heart Failure	4/43 (10%)
Coronary Artery Disease	5/43 (12%)
Hyperlipidemia	20/43 (47%)
Smoker	34/43 (79%)
Previous Stroke	3/43 (7%)
Antiplatelet Use	23/43 (53%)
Ruptured Aneurysm	2/43 (5%)
Aneurysm Location	
MCA	11/43 (26%)
ICA	3/43 (7%)
BA	12/43 (28%)
ACA	2/43 (5%)
ACOMM	13/43 (30%)
PCOMM	1/43 (2%)
VA	1/43 (2%)
Total WEBS Used	
SL	49/50 (98%)
SLS	1/50 (2%)
Number of Second Attempts	7/50 (14%)
Median Device Volume *	113.1 mm^3^
Median Aneurysm Volumes	
3D Automatic	151.4 mm^3^
3D Manual **	108.2 mm^3^
2D DSA **	127.5 mm^3^

MCA, middle cerebral artery; ICA, internal carotid artery; BA, basilar artery; ACA, anterior cerebral artery; ACOMM, anterior communicating artery; PCOMM, posterior communicating artery; VA, vertebral artery; SL, single layer; SLS, single layer sphere. * Calculated with formula πr^2^h (SL) and 4/3 πr^3^ (SLS). ** Calculated with formula (π/6 × width × depth × height).

## Data Availability

Data that supports the findings of this study are available from the corresponding author upon reasonable request.

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
