# Peer review of "Optimal Woven EndoBridge (WEB) Device Size Selection Using Automated Volumetric Software"

_brainsci, 2021, doi:10.3390/brainsci11070901_

Round 1

Reviewer 1 Report

I read your manuscript with great interest. Below are some points which you might want to discuss further:

1- Did you notice any differences when comparing small aneurysms with larger aneurysms in your cohort in terms of DAV ratios and reliability of the measurements?

2- The two aneurysms in the figures have both smooth margins without lobulations. In aneurysms with irregular shape and daughter sacs, sometimes only the proximal part or just one lobulation is intentionally occluded with the WEB device in order to treat the aneurysm. Did you have such aneurysms in your cohort and could you comment on that?     

Author Response

Reply to Reviewer Comments

Reviewer #1:

  1. Did you notice any differences when comparing small aneurysms with larger aneurysms in your cohort in terms of DAV ratios and reliability of the measurements?
    1. Response: Thank you for the comment. A sensitivity analyses showed no differences in the DAV ratios among the different aneurysm sizes,  regarding their implantation status. We have added the results and the discussion in the revised manuscript. (Methods section 2.4; Page 5, paragraph 2, Result section; page 8 and discussion section; page 9)
  2. The two aneurysms in the figures have both smooth margins without lobulations. In aneurysms with irregular shape and daughter sacs, sometimes only the proximal part or just one lobulation is intentionally occluded with the WEB device in order to treat the aneurysm. Did you have such aneurysms in your cohort and could you comment on that?  
    1. Response: Thank you for the thoughtful comment by the reviewer. Most of the aneurysms in our cohort had an ellipsoid/spherical shape, only 8 had irregular margins with 2 or more daughter sacs/lobes. We have added the description and the discussion in the revised manuscript. (Method section 2.4; page 5, result section page 8 and discussion section pages 9-10)

Again, thank you for the reviewer for pointing it out, we really appreciate it. All the changes are reflected in track changes, and the manuscript now looks better.

Reviewer 2 Report

This is a well done paper. I have no corrections. 

Author Response

Reviewer 2 had no comments or suggestions.